# Method Validation for Multi-Pesticide Residue Determination in Chrysanthemum

**DOI:** 10.3390/molecules28031291

**Published:** 2023-01-29

**Authors:** Xinru Wang, Zihan Wang, Jiawei Yu, Luchao Wu, Xinzhong Zhang, Fengjian Luo, Zongmao Chen, Li Zhou

**Affiliations:** 1Tea Research Institute, Chinese Academy of Agricultural Sciences, Hangzhou 310008, China; 2Key Laboratory of Biology, Genetics and Breeding of Special Economic Animals and Plants, Ministry of Agriculture and Rural Affairs, Hangzhou 310008, China

**Keywords:** pesticide residue, analytical method, consumable flowers, chrysanthemum

## Abstract

The chrysanthemum can be consumed in various forms, representing the “integration of medicine and food”. Quantitative analysis of multi-pesticide residues in chrysanthemum matrices is therefore crucial for both product-safety assurance and consumer-risk evaluation. In the present study, a simple and effective method was developed for simultaneously detecting 15 pesticides frequently used in chrysanthemum cultivation in three matrices, including fresh flowers, dry chrysanthemum tea, and infusions. The calibration curves for the pesticides were linear in the 0.01–1 mg kg^−1^ range, with correlation coefficients greater than 0.99. The limits of quantification (LOQs) for fresh flowers, dry chrysanthemum tea, and infusions were 0.01–0.05 mg kg^−1^, 0.05 mg kg^−1^, and 0.001–0.005 mg L^−1^, respectively. In all selected matrices, satisfactory accuracy and precision were achieved, with recoveries ranging from 75.7 to 118.2% and relative standard deviations (RSDs) less than 20%. The validated method was then used to routinely monitor pesticide residues in 50 commercial chrysanthemum-tea samples. As a result, 56% of samples were detected with 5–13 pesticides. This research presents a method for the efficient analysis of multi-pesticide residues in chrysanthemum matrices.

## 1. Introduction

Herbs have long been utilised in both Eastern and Western cultures as medicine and food additives. According to a WHO survey, almost 70–80% of the world’s population, particularly in developing countries, uses non-conventional medicine, primarily from herbal sources, for primary health care [1]. Chinese Herbal Medicines (CHMs) are currently widely utilised around the world, and their popularity is growing [2]. As one of the most prevalent CHMs, chrysanthemum (*Dendranthema grandilora*) has been broadly applied as a heat-clearing and detoxifying herb. Chrysanthemum extract has also been found to produce potent anti-oxidative stress, anti-inflammatory, and anti-tumour effects, etc. [3,4]. In addition, chrysanthemum-tea is a popular healthy beverage, not only due to its distinctive taste and flavour, but also because it contains necessary dietary compounds such as amino acids, vitamins, and trace elements [5]. Therefore, as a good example of “integration of medicine and food”, the chrysanthemum has a large consumer group [6].

In order to meet the rising demand for chrysanthemum products, the cultivation area for chrysanthemum has expanded rapidly over recent years [7]. Meanwhile, a great range of pesticides were used in chrysanthemum cultivation to preserve the chrysanthemum crops and to prevent loss from pests and diseases such as aphids*, Phytoecia rufiventris Gautier, prodenia litura,* thrips, mosaic disease, grey mould, and rust. In China, over 90 pesticide products have been registered on chrysanthemum to date. In tandem with the widespread and excessive use of pesticides, public concern regarding the safety of chrysanthemum products has increased. The frequent presence of pesticide residues in chrysanthemum products poses a potential threat to consumer health. In reality, chrysanthemum teas probably contain higher levels of pesticide residues than green and black teas, due primarily to the simplicity of their production [8]. For the majority of chrysanthemum teas, only drying is required; other traditional manufacturing steps (such as fermentation, rolling, and heating) that may result in pesticide loss are unnecessary [9,10]. Pesticide residues in relatively high levels have been reported in commercial chrysanthemum flowers as well as chrysanthemum teas [11,12,13]. For example, according to a profiling study by Feng et al., a total of 25 pesticides were detected, with detection rates above 10% in 75 chrysanthemum tea samples, with carbendazim (with a detection rate of 91%), bifenthrin (68%), and imidacloprid (63%) being the top three pesticides [14]. Thus, monitoring of pesticide residues in chrysanthemum products has become essential for their safety evaluation. While chrysanthemum infusion is the most common method of consumption, either by personal brewing or industry processing as ready-to-drink beverages, quantitative analysis of multi-pesticide residues in chrysanthemum infusion is also of critical importance for risk assessment for consumers.

Sample pretreatment is critical for accurate quantification of the target analytes. The most commonly used sample pretreatment methods for pesticide residue analysis include solid-phase extraction (SPE), liquid–liquid extraction (LLE), gel permeation chromatography (GPC), solid-phase micro-extraction (SPME) and QuEChERS [15,16,17,18]. Since developed by Anastassiades et al. in 2003 [19], QuEChERS (quick, easy, cheap, effective, rugged, and safe) concepts have been rapidly developed and extensively used for analytical applications of pesticide, clinical, veterinary drug, environmental pollutants, and so on [20,21,22]. Currently, QuEChERS is the first choice for rapid pretreatment in a wide variety of substrates [23,24]. Wang et al. established a sin-QuEChERS coupled with GC-Orbitrap-MS method for simultaneous determination of 352 pesticide residues in chrysanthemum [6]. Fan et al. established a modified QuEChERS method in combination with HPLC-MS/MS for simultaneous determination of 108 pesticides in three traditional Chinese medicines, including *F. thunbergii*, *C. morifolium*, and *D. officinale* [25]. 

In the present study, a simple, fast, and effective approach for detecting 15 pesticides commonly used in chrysanthemum cultivation in three chrysanthemum matrices, consisting of fresh chrysanthemum flowers, dry chrysanthemum tea, and chrysanthemum infusions, was established. The method, demonstrated with satisfactory accuracy and precision, was then successfully applied to accurately analyse the pesticide residues in market samples. This study provides reliable methodological support for the safety evaluation of chrysanthemum products.

## 2. Results and Discussion

### 2.1. Optimisation of the Instrument Conditions 

In addition to high selectivity and sensitivity, triple–quadrupole mass spectrometers provide practical solutions for simultaneous determination of multiple pesticide residues [26,27]. For method-establishment of pesticides in group 1 using UPLC-MS/MS, the single standard solution of 1 mg L*^−^*^1^ for each pesticide was directly infused into the ESI source at 10 µL min*^−^*^1^ to acquire the parent ions, daughter ions, as well as the corresponding optimised cone voltage and collision energy (CE). Consequently, all of the compounds exhibited greater intensity in the positive electrospray ionisation mode (ESI^+^). Two MRM transitions were finally applied for each analyte, one for quantitation and another for confirmation (Table 1). The overlay MRM chromatograms of the 12 pesticides at 1 mg L*^−^*^1^, acquired under the optimal MRM conditions, were shown in Figure 1A. For the GC-amenable pesticides in group 2, MRM mode was used for quantification during GC-MS/MS acquisition. The two MRM transitions for each compound are shown in Table 1, the optimum CEs for the six pesticides varied from 5 to 36 eV. Figure 1B shows the MRM chromatograms of the six pesticides with parameters in optimisation.

### 2.2. Optimisation of Sample Extraction and Clean-Up Procedure

In this study, an optimisation experiment of sample extraction and clean-up was developed in order to establish a simple and economical method with high efficiency. To begin, acetonitrile (MeCN) and 1% formic acid (FA) in MeCN (1% FA-MeCN) were applied as extraction solvents in two groups of recovery experiments, respectively. As a result (Figure 2A), almost all of the analytes presented better recoveries when MeCN rather than acidified MeCN was used, so MeCN was chosen as the extraction solvent. The experiment on recovery rates caused by different MeCN extraction times was then carried out. As shown in Figure 2B, some pesticides were identified with substantially higher recoveries (*p* < 0.05) by double extraction than by one-time extraction, but one-time extraction was sufficient to meet the criteria for accuracy (recoveries between 76.1 and 86.6%) and precision (RSDs between 3.8 and 14.5%). Therefore, in order to achieve targets including quick, easy, cheap, effective, and green, one-time extraction with MeCN was used for sample extraction in this study. 

SPE strategy with TPT cartridge and a modified QuEChERS strategy with different GCB dosages (30 mg and 50 mg) were compared to optimise the cleaning procedure. As shown in Figure 3, all analytes obtained satisfactory recoveries (≥75.0%) using both SPE and modified QuEChERS methods, except for tebuconazole, which had a recovery rate of 64.4% with the modified QuEChERS using 30 mg GCB. Furthermore, due to the high cost of the TPT cartridge, a modified QuEChERS method with 1.8 g NaAC, 100 mg PSA, and 50 mg GCB was eventually used.

### 2.3. Method Validation

#### 2.3.1. Linearity, Matrix Effects and Limit of Detection

Matrix effect (ME) was typically caused by interference from co-extraction and might compete with a component during the ionisation process at the ion source [28]. Overall, ME was computed using the slope of the matrix-matched calibration standards and the solvent standards at the same gradient concentrations, as defined by Equation (1). In general, the ME was rated as low between −20% and 20%, moderate between −50% and 50%, and strong between <−50% and >50% [29]. As shown in Table 2, excellent linearity (R^2^ (correlation coefficient) ≥0.9909) was obtained for all pesticides within the concentration range of 0.01–1 mg L*^−^*^1^ in each of the three experimental matrices. As shown in Figure 4, nearly all matrices exhibited matrix suppression effects for pesticides in group 1 by UPLC-MS/MS analysis, whereas matrix improvement effects were observed in all matrices for pesticides in group 2 by GC-MS/MS analysis. Moreover, the dry chrysanthemum-tea samples exhibited the strongest MEs for the majority of pesticides, which may have been caused by the concentration of the complex matrix components during chrysanthemum processing. The limit of detection (LOD) was defined by a signal-to-noise (S/N) ratio of three at the lowest linear range concentration levels in matrix solvent. Therefore, the LODs ranged from 0.0004 to 0.018 mg kg*^−^*^1^.

#### 2.3.2. The Accuracy, Precision, and Limit of Quantification

To assess the accuracy and precision of the developed method, recovery tests with three spike levels (*n* = 5) were used to validate the method. According to Table 3, the recoveries of the 15 pesticides were ranged from 75.7 to 118.2%, with relative standard deviations (RSDs) ranging from 0.6% to 19.2% in all chosen matrices. The LOQ was defined as the lowest spike level in recovery experiments. In this instance, the LOQs were determined at 0.05 mg kg*^−^*^1^ for all pesticides in dry chrysanthemum tea samples, although in fresh flowers (and infusions), the LOQs were 0.01 mg kg*^−^*^1^ (0.001 mg L*^−^*^1^ for infusions) and 0.05 mg kg*^−^*^1^ (0.005 mg L*^−^*^1^ for chrysanthemum infusions) for the LC- and GC-amenable pesticides, respectively. 

### 2.4. Method Application

In order to confirm the efficacy of the established method and its suitability in routine analysis, the method was applied to fifty chrysanthemum samples from a market. Pesticide residues were detected in 45 of the samples. Table 4 summarises the pesticides monitored, with detected frequencies ranging from 4.0 to 64.0%. Only one of the 15 pesticides, imidacloprid, is subject to regulation in China, with an MRL of 2 mg kg*^−^*^1^ on dry chrysanthemum and an MRL of 2 mg kg*^−^*^1^ on fresh chrysanthemum for thiamethoxam. Therefore, in the absence of relevant information, the MRLs for pesticides in conventional tea were referenced. In that case, this batch of samples was tested with exceeding frequencies of 0–8.0%. Moreover, 56% of samples were found to contain multiple pesticide residues ranging from 5 to 13. Thus, the formulation of MRL criteria for pesticide residues in chrysanthemums is urgently required for compliance and guidance of proper pesticide application on chrysanthemums, as well as to assure food safety.

## 3. Materials and Methods

### 3.1. Materials and Reagents

Pesticide standards including imidacloprid, acetamiprid, thiamethoxam, dinotefuran, pyraclostrobin, dimethomorph, tebuconazole, difenoconazole, clothianidin, bifenthrin, cypermethrin, cyhalothrin, chlorfenapyr, deltamethrin, and chlorpyrifos were purchased from Anpu Chemistry CO., Ltd. (Shanghai, China). The standard stock solution of 100 mg L^−1^ of mixture pesticides was configured in acetonitrile (MeCN) and kept at −20 °C. The working standard solutions were prepared freshly. HPLC grade MeCN, methanol (MeOH) and hexane were purchased from Fisher Scientific CO., Ltd. (Shanghai, China). HPLC benzene was from Yonghua Chemical CO., Ltd. (Jiangsu, China). HPLC acetone was from Tedia Company, Inc. (Cincinnati, OH, USA). HPLC ammonium acetate (≥98.0%) was obtained from ANPEL Laboratory Technologies (Shanghai, China). HPLC formic acid (FA, ≥99.0%) was purchased from Macklin Biochemical CO., Ltd. (Shanghai, China). Magnesium sulfate anhydrous (MgSO_4_, AR, ≥98.0%) and sodium acetate anhydrous (NaAC, AR, ≥99.0%) were purchased from Lingfeng Chemistry Reagent Co., Ltd. (Shanghai, China). Sodium chloride (NaCl, AR, ≥99.5%) was obtained from Guanghua Sci-Tech Co., Ltd. (Guangdong, China). Graphitised carbon black (GCB, 120–400 mesh) and primary–secondary amine (PSA, 40–63 um) were obtained from Bonna Agela Technologies Co., Ltd. (Tianjin, China). Flolisil was purchased from Jixiang Chemical CO., Ltd. (Zhejiang, China).

### 3.2. Sample Preparations

For method development, blank samples of fresh chrysanthemums were obtained from a chrysanthemum field in Tongxiang (30.37°N, 120.28°E, Zhejiang, China). The dry chrysanthemum tea samples were made from fresh flowers by steaming (105 °C, 2 min) and then drying (60 °C, 5 h). The infusion was prepared by brewing the dry chrysanthemum tea in boiling water at a 1:50 ratio for 5 min.

The pretreatment for pesticides in group 1 (Table 1) in chrysanthemum samples was listed as follows: 2.0 g ground dry chrysanthemum tea samples (4.0 g of fresh flower samples) were weighed into a 50 mL centrifuge tube and soaked in 10 mL of water (H_2_O) for 30 min. Then 20 mL MeCN was added, and the mixture was vortexed for 3 min before being centrifuged for 5 min at 5000 r min^−1^. Following that, a 10 mL aliquot of the supernatant was transferred into a 30 mL centrifuge tube containing 1.8 g NaAC, 100 mg PSA, and 50 mg GCB. The extract was then vortexed for 2 min before being centrifuged for 5 min at a rotary speed of 10000 r min^−1^. Finally, 8.0 mL of the upper layer was evaporated to near dryness, redissolved in 1 mL MeOH, and filtered through an organic membrane (0.22 µm) prior to UPLC-MS/MS analysis. For infusion samples, a 20 mL chrysanthemum infusion sample was placed in a 50 mL centrifuge tube, and 20 mL MeCN was added. After vortexing for 3 min, 4.0 g NaCl, 2.0 g MgSO_4_, and 1.5 g NaAC were added. The extract was vortexed and centrifuged for 5 min at 5000 r min^−1^. Following that, 12 mL of the upper layer was transferred into a 30 mL centrifuge tube containing 1.2 g NaAC and 100 mg PSA. An aliquot of 10 mL of supernatant was dried after being vortexed and centrifuged. The residue was then redissolved in 1 mL MeOH, and the sample was filtered through an organic membrane (0.22 µm) prior to UPLC-MS/MS analysis.

For pesticides in group 2 (Table 1), 2.0 g of ground dry-chrysanthemum tea samples (4.0 g of fresh samples) was weighed in a 50 mL centrifuge tube, and 20 mL of hexane/acetone (1/1) was applied for extraction. The mixture was then vortexed for 3 min and centrifuged at 5,000 r min^−1^ for 5 min. An aliquot of 10 mL of supernatant was then evaporated to dryness and redissolved in 2 mL of a solvent mixture containing hexane/acetone/benzene (440/10/50). After that, a glass column filled with 2.0 g florisil and 0.05 g GCB was prepared, and the column was prewashed with hexane/acetone/benzene (440/10/50) prior to sample loading, then a total of 10 mL of elution was collected by eluting with hexane/acetone/benzene (440/10/50). Then 5 mL of the eluent was concentrated to dryness, redissolved in 1 mL MeCN, and filtered for GC-MS/MS analysis. In the case of infusion samples, a 10 mL chrysanthemum infusion was measured into a 50 mL centrifuge tube and extracted with 20 mL hexane. After vortexing, 3.0 g NaCl was added, and then the upper hexane was transferred after centrifugation. The extraction process was then repeated a second time, and the two extracts were mixed. Finally, the mixture was evaporated to dryness, redissolved in 1 mL MeCN, and filtered prior to GC-MS/MS analysis.

### 3.3. UPLC-MS/MS Analytical Conditions

The 9 pesticides in Group 1 were analysed using UPLC-MS/MS. A Waters Acquity UPLC system in tandem with a Waters Xevo TQ-S Micro triple-quadrupole mass spectrometer (Waters, Milford, MA, USA) equipped with an Electrospray Ionisation (ESI) source was performed. Chromatography was performed at 40 °C on an ACQUITY UPLC HSS T3 column (100 mm × 2.1 mm, 1.8 m; Waters, Milford, MA, USA). The mobile phase consisted of 0.1% formic acid in MeOH (A) and 10 mmol L^−1^ ammonium acetate (B). The separation was run at a flow rate of 0.2 mL min-1 with the following elution gradient: 1.0% A initially, 50% A for 0–4 min, 90% A for 4.0–5.2 min, 100% A for 5.2–5.3 min (held for 2.7 min), and then 1.0% A for 7.0–7.6 min. The total running time was 12 min, with an injection volume of 5 µL. Table 1 shows the scheduled MRM parameters for the pesticides.

### 3.4. GC-MS/MS Analytical Conditions

An Agilent 8890 gas chromatograph coupled to an Agilent 7000D GC/TQ mass spectrometer (Agilent, Stevens Creek, CA, USA) was used for pesticide analysis in group 2. Separation was developed on an Agilent J&W HP-5ms GC Column (30 m × 0.25 mm × 0.25 µm), with the oven temperature set as follows: 40 °C for 1 min, then raised to 120 °C by 40 °C min^−1^, then to 240 °C by 5 °C min^−1^, then to 300 °C (final temperature) at 12 °C min^−1^ and maintained for 5 min. Helium (>99.999%) was used as the carrier gas, with a constant flow rate of 2.25 mL min^−1^, and nitrogen (>99.999%) was used as the collision gas. The ion source and transfer line were both set to 280 °C. The injection volume was 1.0 µL. The electron energy was 70 eV. MRM mode was used in the mass spectrometer analysis of target compounds (Table 1).

### 3.5. Method Validation

The blank matrices, including fresh flowers, dry tea samples, and chrysanthemum infusions, were prepared as described in Section 2.2 and were used for validation. Method validation was assessed based on linearity, precision, accuracy, LOQs, and ME [30]. For linearity estimation, the matrix-matched calibration with gradient concentrations (0.01–1 mg L^−1^) was utilised. Recovery assays with three spiked levels and five replicates were conducted to evaluate the accuracy and precision of the method. To meet the demands of accuracy and precision in recovery experiments, LOQs for pesticides were determined as the minimum spiked concentration. The ME was calculated using the following equation [31,32]:ME (%) = (A/B − 1) × 100(1)
where A and B represented the slope of the matrix-matched and solvent calibration curves, respectively.

## 4. Conclusions

A method for simultaneously detecting 15 pesticides commonly used in chrysanthemum cultivation was established on three chrysanthemum matrices, including fresh flowers, dry tea, and infusions. The approach was proven to be reliable because the validation results obtained from the three matrices indicated good linearity, accuracy, and precision. The LOQs were 0.01–0.05 mg kg^−1^, 0.05 mg kg^−1^ and 0.001–0.005 mg L^−1^ for fresh flower, dry chrysanthemum tea and infusions, respectively. Following that, the method was subsequently applied to monitor pesticide residues in 50 commercial chrysanthemum teas. With the benefits of simple and economic pretreatment as well as accurate analysis, the developed method provides technical support for efficient analysis of pesticide residues in chrysanthemum matrices, as well as other similar CHMs and related products in the future.

## Figures and Tables

**Figure 1 molecules-28-01291-f001:**
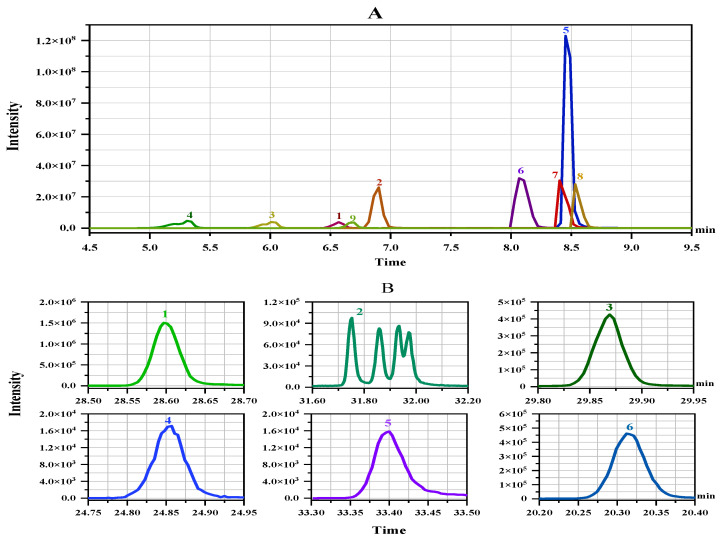
The representative MRM chromatograms of the 15 pesticides by LC-MS/MS (**A**) and GC-MS/MS (**B**) (The numbers correspond to those in Table 1).

**Figure 2 molecules-28-01291-f002:**
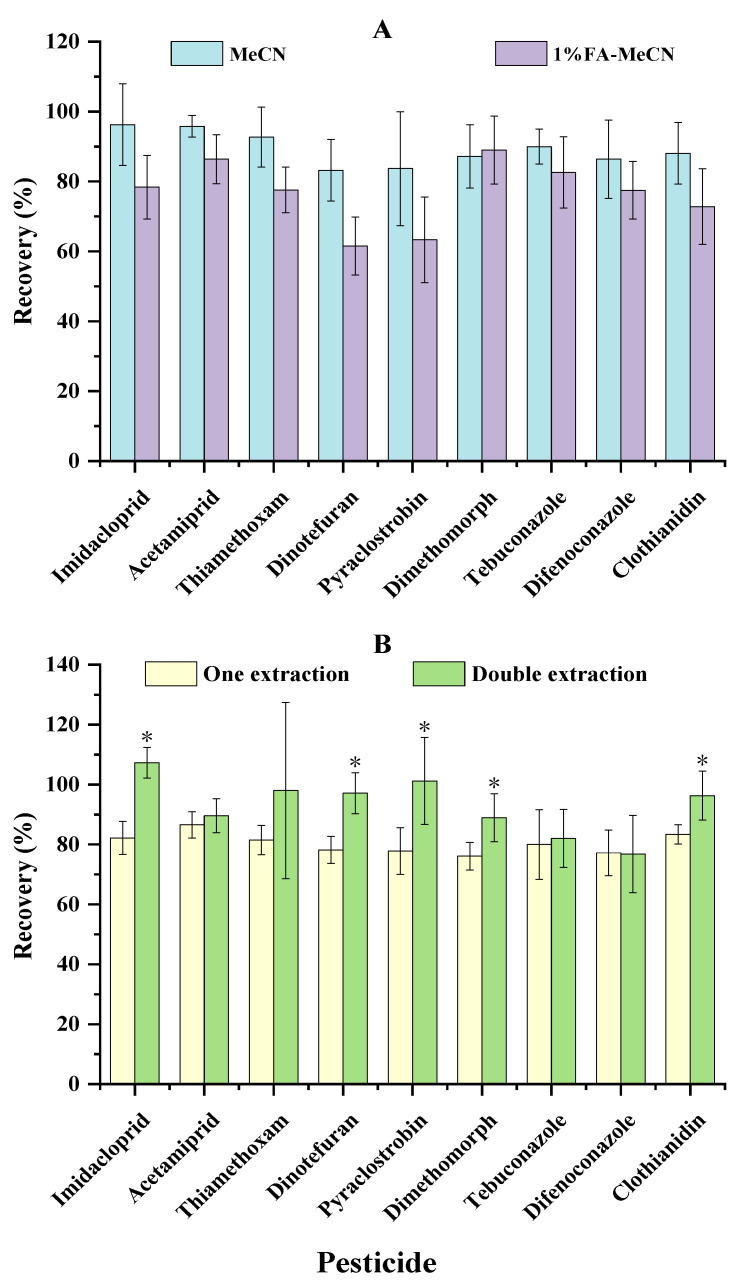
The recoveries of the LC-amenable pesticides with different extraction solvents (**A**) and times (**B**) (*presents *p* < 0.05).

**Figure 3 molecules-28-01291-f003:**
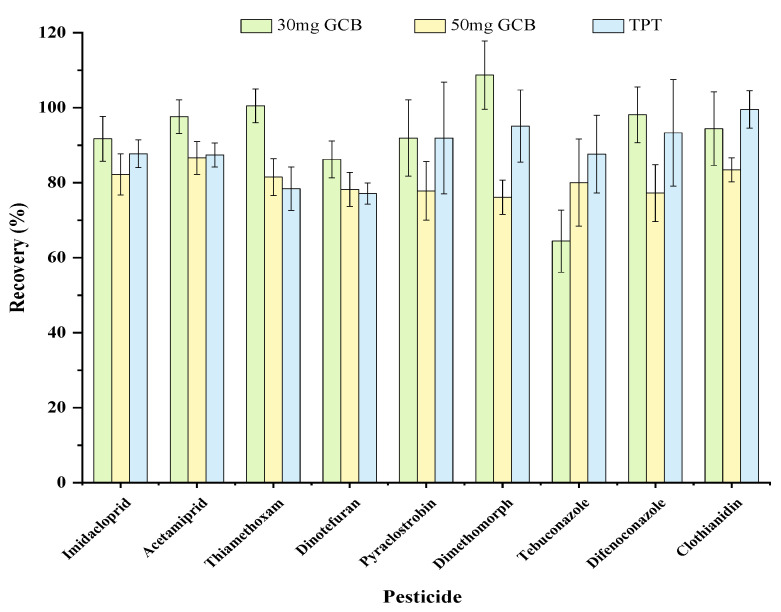
The recoveries of the LC-amenable pesticides with different clean-up procedures of TPT cartridge and a modified QuEChERS strategy.

**Figure 4 molecules-28-01291-f004:**
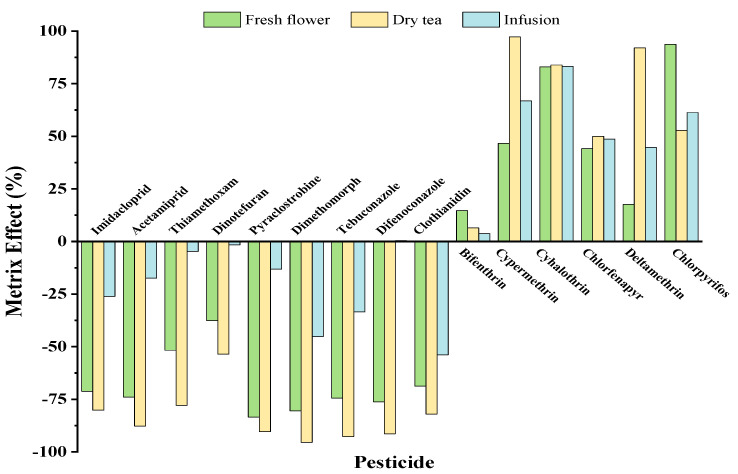
The matrix effect (%) of the 15 pesticides in different chrysanthemum matrices of fresh flower, dry tea and infusion samples.

**Table 1 molecules-28-01291-t001:** The optimised UPLC-MS/MS and GC-MS/MS parameters of Retention time, Cone voltage, Ion transitions and collision energy (CE) for the 15 pesticides.

UPLC-MS/MS								
Group No.	No.	Pesticide	Retention Time	Cone voltage	Quantification		Confirmation	
			(min)	(V)	Ion transitions (*m/z*)	CE/V	Ion transitions (*m/z*)	CE/V
1	1	Imidacloprid	6.57	28	256 > 175	20	256 > 209	15
	2	Acetamiprid	6.90	30	223 > 126	20	223 > 56	15
	3	Thiamethoxam	6.01	20	292 > 211	12	292 > 181	28
	4	Dinotefuran	5.31	15	203 > 129	12	203 > 157	8
	5	Pyraclostrobin	8.45	25	388 > 194	10	388 > 163	20
	6	Dimethomorph	8.07	35	388 > 301	20	388 > 164	40
	7	Tebuconazole	8.40	32	308 > 70	25	308 > 125	35
	8	Difenoconazole	8.53	35	406 > 251	30	406 > 337	15
	9	Clothianidin	6.70	15	250 > 169	15	250 > 132	20
**GC-MS/MS**								
**Group No.**	**No.**	**Pesticide**	**Retention time**		**Quantification**		**Confirmation**	
			**(min)**		**Ion transitions** **(*m/z*)**	**CE/eV**	**Ion transitions (*m/z*)**	**CE/eV**
2	1	Bifenthrin	28.60		181.2 > 165.2	25	181.2 > 166.2	10
	2	Cypermethrin	31.75		181.0 > 152.1	25	163.0 > 91.0	10
	3	Cyhalothrin	29.87		181.1 > 152.0	25	197.0 > 141.0	10
	4	Chlorfenapyr	24.84		362.6 > 247.0	30	362.6 > 308.0	36
	5	Deltamethrin	33.40		250.7 > 172.0	5	181.0 > 152.1	25
	6	Chlorpyrifos	20.32		196.6 > 169.0	15	198.9 > 171.0	15

**Table 2 molecules-28-01291-t002:** The Linear range, Regression equations, Correlation coefficient (R^2^), Matrix effects (MEs) and Limits of detection (LODs) of the 15 pesticides in different chrysanthemum matrices.

Pesticide	Matrix	Linear Range (mg L^−1^)	Regression Equation	R^2^	MEs	LODs (mg kg^−1^, mg L^−1^)
Imidacloprid	Fresh flower	0.01–1	y = 139,567x + 2881	0.9909	−71.2	0.006
	Dry tea		y = 72,481x + 2032	0.9990	−80.2	
	Infusion		y = 289,810x + 7562	0.9988	−26.1	
Acetamiprid	Fresh flower	0.01–1	y = 781,946x + 26,204	0.9951	−73.9	0.0004
	Dry tea		y = 368,802x + 18,433	0.9999	−87.7	
	Infusion		y = 2,018,129x + 66,236	0.9911	−17.4	
Thiamethoxam	Fresh flower	0.01–1	y = 330,845x + 22,513	0.9974	−51.7	0.002
	Dry tea		y = 139,527x + 1150	0.9987	−77.9	
	Infusion		y = 647,351x + 12,317	0.9985	−4.7	
Dinotefuran	Fresh flower	0.01–1	y = 601,655x + 23,365	0.9970	−37.6	0.0009
	Dry tea		y = 427,799x + 14,629	0.9988	−53.5	
	Infusion		y = 816,026x + 54,288	0.9915	−1.6	
Pyraclostrobin	Fresh flower	0.01–1	y = 867,185x + 5502	0.9958	−83.5	0.01
	Dry tea		y = 95,703x + 2706	0.9983	−90.4	
	Infusion		y = 9,628,154x − 39,671	0.9938	−13.2	
Dimethomorph	Fresh flower	0.01–1	y = 778,443x + 13,553	0.9917	−80.5	0.002
	Dry tea		y = 226,589x + 3247	0.9977	−95.5	
	Infusion		y = 2,268,069x + 8124	0.9994	−45.2	
Tebuconazole	Fresh flower	0.01–1	y = 510,990x + 1532	1.0000	−74.5	0.01
	Dry tea		y = 73,211x − 1524	0.9997	−92.7	
	Infusion		y = 1,748,172x + 8566	0.9997	−33.5	
Difenoconazole	Fresh flower	0.01–1	y = 238,114x + 1088	0.9992	−76.2	0.018
	Dry tea		y = 68,335x + 1954	0.9913	−91.4	
	Infusion		y = 2,051,773x + 1066	09989	0.4	
Clothianidin	Fresh flower	0.01–1	y = 114,447x − 1445	0.9977	−68.8	0.003
	Dry tea		y = 72,440x + 39,324	0.9963	−82.0	
	Infusion		y = 160,687x + 273,688	0.9948	−53.9	
Bifenthrin	Fresh flower	0.01–1	y = 7,594,842x − 97,294	0.9954	14.7	0.003
	Dry tea		y = 7,049,790x − 87,718	0.9954	6.5	
	Infusion		y = 6,873,310x + 14,630	1.0000	3.8	
Cypermethrin	Fresh flower	0.01–1	y = 1,231,236x + 622	0.9991	46.6	0.04
	Dry tea		y = 165,641x − 247	0.9998	97.3	
	Infusion		y = 140,052x − 298	1.0000	66.8	
Cyhalothrin	Fresh flower	0.01–1	y = 773,949x − 4788	0.9998	83.0	0.01
	Dry tea		y = 777,851x + 358	0.9999	83.9	
	Infusion		y = 774,970x − 5280	0.9997	83.2	
Chlorfenapyr	Fresh flower	0.01–1	y = 47,735x − 918	0.9948	44.1	0.01
	Dry tea		y = 49,690x − 985	0.9911	50.0	
	Infusion		y = 49,240x − 644	0.9983	48.7	
Deltamethrin	Fresh flower	0.01–1	y = 22,081x − 19	1.0000	17.6	0.012
	Dry tea		y = 36,047x + 33	1.0000	92.0	
	Infusion		y = 27,163x − 116	0.9998	44.7	
Chlorpyrifos	Fresh flower	0.01–1	y = 1,276,562x + 5532	0.9999	93.7	0.004
	Dry tea		y = 1,006,677x + 12,749	0.9994	52.8	
	Infusion		y = 1,062,913x + 689	0.9999	61.3	

**Table 3 molecules-28-01291-t003:** The Average recoveries (AR, %, *n* = 5), Relative standard deviations (RSDs, %) and Limit of quantification (LOQs) of the 15 pesticides in different chrysanthemum matrices at three spiked levels (SLs).

Pesticide	Fresh Flower				Dry Tea				Infusion			
	SL (mg kg^−1^)	AR (%)	RSD (%)	LOQ	SL (mg kg^−1^)	AR (%)	RSD (%)	LOQ	SL (mg L^−1^)	AR (%)	RSD (%)	LOQ
Imidacloprid	0.01	84.3	11.5	0.01	0.05	92.8	12.3	0.05	0.001	95.2	16.6	0.001
	0.1	93.4	10.5		0.1	91.8	8.2		0.01	95.4	7.1	
	1	100.8	8.5		1	90.9	7.4		0.1	91.4	6.9	
Acetamiprid	0.01	88.3	3.4	0.01	0.05	96.2	3.6	0.05	0.001	93.8	10.8	0.001
	0.1	90.3	6.4		0.1	99.7	3.7		0.01	98.2	3.0	
	1	93.4	8.2		1	89.9	5.5		0.1	96.2	4.7	
Thiamethoxam	0.01	89.7	6.0	0.01	0.05	93.7	10.8	0.05	0.001	95.0	15.3	0.001
	0.1	105.1	5.7		0.1	95.7	4.7		0.01	91.1	5.3	
	1	97.3	10.0		1	83.8	4.2		0.1	92.0	4.8	
Dinotefuran	0.01	75.9	2.2	0.01	0.05	85.4	4.3	0.05	0.001	89.6	12.5	0.001
	0.1	88.5	8.3		0.1	92.9	6.8		0.01	87.5	6.4	
	1	79.0	5.1		1	82.2	6.4		0.1	86.2	6.7	
Pyraclostrobin	0.01	93.3	9.4	0.01	0.05	79.4	5.0	0.05	0.001	101.5	7.0	0.001
	0.1	101.1	7.8		0.1	93.6	15.7		0.01	97.8	4.0	
	1	93.2	6.3		1	78.5	3.1		0.1	96.2	3.8	
Dimethomorph	0.01	96.5	10.7	0.01	0.05	87.0	9.4	0.05	0.001	92.4	11.3	0.001
	0.1	90.8	11.7		0.1	97.7	8.3		0.01	99.0	3.8	
	1	92.6	3.9		1	75.7	12.1		0.1	99.7	1.8	
Tebuconazole	0.01	86.1	15.2	0.01	0.05	106.6	18.2	0.05	0.001	94.3	4.3	0.001
	0.1	100.0	2.6		0.1	88.2	15.7		0.01	102.2	5.7	
	1	92.8	6.1		1	97.3	13.5		0.1	97.7	2.8	
Difenoconazole	0.01	103.6	3.7	0.01	0.05	102.4	8.3	0.05	0.001	91.8	9.0	0.001
	0.1	102.5	5.0		0.1	97.6	6.5		0.01	102.3	6.0	
	1	93.4	10.1		1	88.4	5.6		0.1	99.2	4.0	
Clothianidin	0.01	81.2	18.8	0.01	0.05	82.0	17.8	0.05	0.001	76.6	12.7	0.001
	0.1	92.2	8.4		0.1	87.7	8.6		0.01	106.1	3.2	
	1	99.6	4.3		1	91.2	18.7		0.1	101.4	4.3	
Bifenthrin	0.05	118.0	0.6	0.05	0.05	94.8	6.3	0.05	0.005	95.3	4.3	0.005
	0.5	84.5	7.7		0.1	94.3	7.5		0.01	107.5	6.2	
	1	90.5	11.3		1	107.4	8.6		0.1	100.2	3.9	
Cypermethrin	0.05	109.2	1.0	0.05	0.05	97.0	1.5	0.05	0.005	90.8	5.7	0.005
	0.5	95.5	8.8		0.1	113.2	8.8		0.01	111.7	5.0	
	1	88.3	13.1		1	111.3	6.9		0.1	88.8	5.9	
Cyhalothrin	0.05	118.2	1.5	0.05	0.05	104.1	6.6	0.05	0.005	88.2	4.5	0.005
	0.5	103.6	19.2		0.1	97.5	4.4		0.01	105.5	6.8	
	1	88.9	15.8		1	108.2	9.8		0.1	97.3	4.3	
Chlorfenapyr	0.05	115.1	2.6	0.05	0.05	88.0	18.8	0.05	0.005	100.7	8.3	0.005
	0.5	77.1	8.6		0.1	95.7	7.2		0.01	109.6	7.8	
	1	85.7	13.6		1	106.8	8.4		0.1	98.1	3.8	
Deltamethrin	0.05	104.9	7.8	0.05	0.05	103.8	12.9	0.05	0.005	78.2	8.0	0.005
	0.5	81.7	18.1		0.1	100.2	7.2		0.01	110.5	8.3	
	1	79.4	6.4		1	90.3	16.2		0.1	86.6	6.7	
Chlorpyrifos	0.05	116.0	2.4	0.05	0.05	98.8	10.8	0.05	0.005	91.8	1.0	0.005
	0.5	79.9	13.8		0.1	96.0	4.6		0.01	107.6	10.2	
	1	87.8	14.2		1	107.7	8.9		0.1	95.8	3.1	

**Table 4 molecules-28-01291-t004:** Pesticide residues in the 50 dry chrysanthemum tea samples purchased at a Chinese market.

Pesticide	CN MRL	Detected Frequencies	Range	Exceeding MRL Frequencies
	(mg kg^−1^)	(%)	(mg kg^−1^)	(%)
Imidacloprid	2	56	<LOQ −2.00	2
Acetamiprid	10 *	56	<LOQ −2.80	0
Thiamethoxam	2	42	<LOQ −3.32	2
Dinotefuran	20 *	4	<LOQ −0.41	0
Pyraclostrobin	10 *	54	<LOQ −6.45	0
Dimethomorph	-	16	<LOQ −3.73	-
Tebuconazole	-	40	<LOQ −3.54	-
Difenoconazole	10 *	46	<LOQ −2.22	0
Clothianidin	10 *	28	<LOQ −0.64	0
Bifenthrin	5 *	56	<LOQ −2.80	0
Cypermethrin	20 *	64	<LOQ −1.37	0
Cyhalothrin	15 *	38	<LOQ −0.41	0
Chlorfenapyr	20 *	28	<LOQ −1.14	0
Deltamethrin	10 *	8	<LOQ −1.40	0
Chlorpyrifos	2 *	32	<LOQ −5.02	8

Note: * denotes that the MRLs are referred to the standards for tea in China (CN).

## Data Availability

Not applicable.

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
