# Peer review of "Method Validation for Multi-Pesticide Residue Determination in Chrysanthemum"

_molecules, 2023, doi:10.3390/molecules28031291_

Round 1
Reviewer 1 Report
Comments: The quality safety supervision and test of agricultural products urgently need proper analytical method to assess, prevent and control pesticide residues. The present study explores the establishment of a method with UPLC-MS/MS and GC-MS/MS for simultaneous determination of 15 pesticides commonly used in chrysanthemum cultivation in three chrysanthemum matrices. From my point of view, the work is impressive and thus it merits to be published in the Molecules with minor revisions.
Overall Recommendations: Accept after Minor Revisions
1. Abstract
(1) All the abbreviations must be fully described at first mention along with brackets.
(2) The test results of the 50 commercial chrysanthemum tea samples deserve a brief description.
2. Introduction
(1) Cite more references in recent five years.
(2) Verify the written of the nouns in L. 44, whether in correct italics?
3. Results and Discussion
(1) The abscissa data of retention time in Figure 1B should be uniformly reserved with two decimal places, consistent with Table 1.
(2) The abbreviations must be fully described at first mention.
(3) Figure 2 indicates the algorithm used to obtain the significant difference analysis result p<0.05, and the "p" should be italicized.
(4) The "dosage" in L. 128 should be plural. There are some similar syntax errors, the readability of the manuscript needs to be improved.
(5) As mentioned above, the three chrysanthemum matrices, including fresh flower, dry chrysanthemum tea and infusion were tested. Unify expression in this section, including text, tables and figures.
(6) In Table 4, the author should indicate what "*" means.
4. Materials and Methods
(1) All manufacturers have (City, State) after mention.
(2) Why does the dry tea and infusion samples of chrysanthemum obtained as described? In reference with any standards or other articles?
Author Response
Dear reviewer,
Thank you very much for your positive comments and useful suggestions on my manuscript entitled ''Method Validation for multi-pesticide residue determination in Chrysanthemum''(molecules-2095650). We have revised our manuscript according to reviewer's comments and suggestions. Please find our itemized responses to these comments below. We wish the revised manuscript is acceptable for publication in the Molecules.
Both the ''revised manuscript with changes marked'' and one copy of “revised manuscript with no changes marked” for publication-ready have been submitted. All revised figures and tables have been submitted in the text and supplementary material.
Once again, thank you very much for your kind help on our manuscript.
Reviewers' Comments:
Reviewer #1: The quality safety supervision and test of agricultural products urgently need proper analytical method to assess, prevent and control pesticide residues. The present study explores the establishment of a method with UPLC-MS/MS and GC-MS/MS for simultaneous determination of 15 pesticides commonly used in chrysanthemum cultivation in three chrysanthemum matrices. From my point of view, the work is impressive and thus it merits to be published in the Molecules with minor revisions.
Overall Recommendations: Accept after Minor Revisions.
Major comments:
- Abstract
(1) All the abbreviations must be fully described at first mention along with brackets.
(2) The test results of the 50 commercial chrysanthemum tea samples deserve a brief description.
Response: Thanks for your value comments very much. We have added the full description of the abbreviations at first mention and the test of the 50 commercial chrysanthemum tea samples briefly in the revised manuscript.
- Introduction
(1) Cite more references in recent five years.
(2) Verify the written of the nouns in L. 44, whether in correct italics?
Response: Thanks for your comments. According to your suggestion, we have read more recent relevant literature and added more new references in the revised manuscript. And we confirmed the writing format by reference and corrected it in the revised manuscript.
- Results and Discussion
(1) The abscissa data of retention time in Figure 1B should be uniformly reserved with two decimal places, consistent with Table 1.
(2) The abbreviations must be fully described at first mention.
(3) Figure 2 indicates the algorithm used to obtain the significant difference analysis result p<0.05, and the "p" should be italicized.
(4) The "dosage" in L. 128 should be plural. There are some similar syntax errors, the readability of the manuscript needs to be improved.
(5) As mentioned above, the three chrysanthemum matrices, including fresh flower, dry chrysanthemum tea and infusion were tested. Unify expression in this section, including text, tables and figures.
(6) In Table 4, the author should indicate what "*" means.
Response: Sorry for my negligence. According to your detailed suggestion on this chapter, we have uniformed the decimal places in Figure 1B, corrected the italic format of "p" in Figure 2 and plural form of the "dosage" in L. 128. As for the problems about the abbreviations, we checked the whole article and added fully description at first mention along with brackets. We have also made unified statement of the three chrysanthemum matrices across the manuscript, including text, tables and figures. And the meaning of "*" was added in the footnote of Table 4. Additionally, the manuscript has been carefully reviewed and edited by one or more of highly qualified, native English speakers at EditSprings, to assure compliance with Anglophone academic standards in terms of style punctuation, grammar and spelling.
- Materials and Methods
(1) All manufacturers have (City, State) after mention.
(2) Why does the dry tea and infusion samples of chrysanthemum obtained as described? In reference with any standards or other articles?
Response:Thank you for pointing out these problems. I have added the missing information of manufacturers. As for the processing steps of dry chrysanthemum tea, they were referred from the production in the sampling area, and the infusion was obtained referring to the methodology for sensory evaluation of tea in China (Chinese National Standard GB/T 23776-2018). We have already added the reference information of the standard in the revised version.
Thanks again for your comments on this manuscript.
Reviewer 2 Report
The presented article deals with multi-pesticide residue determination in chrysanthemum matrices. Methods assumes application of SPE strategy and modified QuEChERS strategy. The presented research is valuable; however, it has several limitation:
· There is previously reported article [10.1080/00032719.2012.726680] were 100 pesticides were determinate with the use of one method GS-MS/MS
· The investigation of sample prep for GS-MS/MS method was not presented
· In fact two methods with two sample preps are required for 15 analytes determination
Thus, the presented article is not suitable for publication in Molecules
Author Response
Dear reviewer,
Thank you very much for your positive comments and useful suggestions on my manuscript entitled ''Method Validation for multi-pesticide residue determination in Chrysanthemum''(molecules-2095650). We have revised our manuscript according to reviewer's comments and suggestions. Please find our itemized responses to these comments below. We wish the revised manuscript is acceptable for publication in the Molecules.
Both the ''revised manuscript with changes marked'' and one copy of “revised manuscript with no changes marked” for publication-ready have been submitted. All revised figures and tables have been submitted in the text and supplementary material.
Once again, thank you very much for your kind help on our manuscript.
Reviewers' Comments:
Reviewer #2: The presented article deals with multi-pesticide residue determination in chrysanthemum matrices. Methods assumes application of SPE strategy and modified QuEChERS strategy. The presented research is valuable; however, it has several limitation:
- There is previously reported article [10.1080/00032719.2012.726680] were 100 pesticides were determinate with the use of one method GS-MS/MS.
- The investigation of sample prep for GS-MS/MS method was not presented.
- In fact two methods with two sample preps are required for 15 analytes determination.
Thus, the presented article is not suitable for publication in Molecules.
Response: We are very grateful to your comments for our manuscript, and please allow me to make a short explain. Tongxiang (30.37â—¦ N, 120.28â—¦ E, Zhejiang, China), our sampling place, is famous for its rich chrysanthemum. But pesticide contamination, induced by various pesticides applied for insects, fungi, weeds, and disease control, has been a long-term trouble for local chrysanthemum planters and food safety regulatory departments. Thus, quantitative analysis of multi-pesticide residues in chrysanthemum matrices is of urgent need for both safety insurance of the products and risk assessment of the consumers. According to our survey on pesticides application and pesticide residue in market chrysanthemum, the 15 pesticides as our manuscript list, were finalized as analytes for our method establishment in the present study. And as the detection results of the 50 commercial chrysanthemum samples shown, the method was of practical significance in chrysanthemum trade not only at Tongxiang, but also all over the country.
Additionally, for the literature you mentioned, I find that the most pesticides are not commonly used in chrysanthemum cultivation. Out the 15 pesticides we focused on, only 6 were concerned in that research. And because of the huge differences of the properties among the 15 analytes, both GC and UPLC was needed for a better detection effect, so the pesticides were divided into two groups.
Thanks for your attention for this explanation and best wishes for you.
Reviewer 3 Report
For Figures 1, 2, 3 and 4: please expand the captions and elaborate briefly the content of each figure in the caption.
Line 11: Please rewrite the sentence, not able to comprehend its meaning.
Scientific literature review is poorly conducted, needs to be updated, for instance the introduction, results and discussion need to cover latest studies and discoveries on pesticide analysis. I recommend authors to consider some bioanalytical and biosensing-based studies for pesticide detection (see https: //doi.org/10.1016/j.teac.2022.e00184), this will help to enrich the literature survey. See below few important discoveries on novel sensors for pesticide analysis from food such as, fenitrothion (https: //doi.org/10.3390/ijms221910846), malathion (https: //doi.org/10.1016/j.biomaterials.2022.121617), fipronil (https:// doi.org/10.1016/j.jhazmat.2021.127939), and diazinon (https:// doi.org/10.31083/j.fbl2703092).
Please provide values for statistical measurements in abstract and figure captions.
Figure 4. The check axis label—correct the spelling to ‘matrix’ not metrix.
Authors could add the significance and key results into the conclusions sections. Need to elaborate conclusions and provide future directions/implications of the study.
Please signify the relation between the method of analysis and sampling procedures in conclusion and abstract.
Overall, the study is significant for pesticide analysis and could be considered for publication after incorporation of suggested revisions.
Author Response
Dear reviewer,
Thank you very much for your positive comments and useful suggestions on my manuscript entitled ''Method Validation for multi-pesticide residue determination in Chrysanthemum''(molecules-2095650). We have revised our manuscript according to reviewer's comments and suggestions. Please find our itemized responses to these comments below. We wish the revised manuscript is acceptable for publication in the Molecules.
Both the ''revised manuscript with changes marked'' and one copy of “revised manuscript with no changes marked” for publication-ready have been submitted. All revised figures and tables have been submitted in the text and supplementary material.
Once again, thank you very much for your kind help on our manuscript.
Reviewers' Comments:
Reviewer #3: Overall, the study is significant for pesticide analysis and could be considered for publication after incorporation of suggested revisions.
Major comments:
- For Figures 1, 2, 3 and 4: please expand the captions and elaborate briefly the content of each figure in the caption.
Response:Thank you for pointing out these problems. According to your suggestion, we have expanded the captions of figures 1, 2, 3 and 4 to express their meaning as clearly as possible.
- Line 11: Please rewrite the sentence, not able to comprehend its meaning.
Response:Thanks for your detailed suggestion very much. I have corrected my expression to clarify the meaning. And the manuscript has been carefully reviewed and edited by one or more of highly qualified, native English speakers at EditSprings, to assure compliance with Anglophone academic standards in terms of style punctuation, grammar and spelling.
- Scientific literature review is poorly conducted, needs to be updated, for instance the introduction, results and discussion need to cover latest studies and discoveries on pesticide analysis. I recommend authors to consider some bioanalytical and biosensing-based studies for pesticide detection (see https: //doi.org/10.1016/j.teac.2022.e00184), this will help to enrich the literature survey. See below few important discoveries on novel sensors for pesticide analysis from food such as, fenitrothion (https: //doi.org/10.3390/ijms221910846), malathion (https: //doi.org/10.1016/j.biomaterials.2022.121617), fipronil (https:// doi.org/10.1016/j.jhazmat.2021.127939), and diazinon (https:// doi.org/10.31083/j.fbl2703092).
Response: We are very grateful for the valuable literature you provided. By reading and referring these articles, we have improved some contents of our manuscript.
- Please provide values for statistical measurements in abstract and figure captions.
Response: Thanks for your useful comments very much. We have enriched the values for statistical measurements according to your comments.
- Figure 4. The check axis label—correct the spelling to ‘matrix’ not metrix.
Response: Sorry for my negligence. We have corrected the spelling to the word as your indication in the revised manuscript.
- Authors could add the significance and key results into the conclusions sections. Need to elaborate conclusions and provide future directions/implications of the study.
Response: Thanks for your valuable comment. Referring to it, we have added the results of the method application for monitoring pesticide residues in 50 commercial chrysanthemum teas to elaborate the practical use of our study. Additionally, according to your advice, some future implications from this study were proposed in the revised conclusion.
- Please signify the relation between the method of analysis and sampling procedures in conclusion and abstract.
Response: We are grateful for your suggestion. Some contents were supplemented to clarify the relation between the method of analysis and sampling procedures in conclusion and abstract in the revised version.
Thanks again for your comments on this manuscript.
Round 2
Reviewer 2 Report
Authors improved the article and it could be accepted in the present form
Reviewer 3 Report
Authors have revised the manuscript satisfactorily.